# Real-time single-cell characterization of the eukaryotic transcription cycle reveals correlations between RNA initiation, elongation, and cleavage

Jonathan Liu[1], Donald Hansen[2], Elizabeth Eck[3], Yang Joon Kim[3], Meghan Turner[3], Simon Alamos[4], Hernan G. Garcia[1,3,5,6]*

1 Department of Physics, University of California at Berkeley, Berkeley, California, United States of America, 2 Institute of Pharmacy and Molecular Biotechnology, University of Heidelberg, Heidelberg, Germany, 3 Biophysics Graduate Group, University of California at Berkeley, Berkeley, California, United States of America, 4 Department of Plant and Microbial Biology, University of California at Berkeley, Berkeley, California, United States of America, 5 Department of Molecular and Cell Biology, University of California at Berkeley, Berkeley, California, United States of America, 6 Institute for Quantitative Biosciences-QB3, University of California at Berkeley, Berkeley, California, United States of America

* hggarcia@berkeley.edu

**Data Availability Statement:** All software is available on GitHub at https://github.com/

## Abstract

The eukaryotic transcription cycle consists of three main steps: initiation, elongation, and cleavage of the nascent RNA transcript. Although each of these steps can be regulated as well as coupled with each other, their *in vivo* dissection has remained challenging because available experimental readouts lack sufficient spatiotemporal resolution to separate the contributions from each of these steps. Here, we describe a novel application of Bayesian inference techniques to simultaneously infer the effective parameters of the transcription cycle in real time and at the single-cell level using a two-color MS2/PP7 reporter gene and the developing fruit fly embryo as a case study. Our method enables detailed investigations into cell-to-cell variability in transcription-cycle parameters as well as single-cell correlations between these parameters. These measurements, combined with theoretical modeling, suggest a substantial variability in the elongation rate of individual RNA polymerase molecules. We further illustrate the power of this technique by uncovering a novel mechanistic connection between RNA polymerase density and nascent RNA cleavage efficiency. Thus, our approach makes it possible to shed light on the regulatory mechanisms in play during each step of the transcription cycle in individual, living cells at high spatiotemporal resolution.

## Author summary

Live cell imaging using fluorescence microscopy provides an exciting way to visualize the transcription cycle in living organisms with great amounts of precision. However, the output of these technologies is often complex and can be hard to interpret. We have

GarciaLab/TranscriptionCycleInference Data is attached as a supplementary zip file S1 Data.

**Funding:** This work was supported by the Burroughs Wellcome Fund Career Award at the Scientific Interface (https://www.bwfund.org/grant-programs/interfaces-science/career-awards-scientific-interface), the Sloan Research Foundation (https://sloan.org/), the Human Frontiers Science Program (https://www.hfsp.org/), the Searle Scholars Program (https://www.searlescholars.net/), the Shurl and Kay Curci Foundation (http://thecurcifoundation.org/), the Hellman Foundation (https://www.hellmanfoundation.org/), the NIH Director's New Innovator Award (DP2 OD024541-01, https://commonfund.nih.gov/newinnovator), and an NSF CAREER Award (1652236, https://www.nsf.gov/funding/pgm_summ.jsp?pims_id=503214) (HGG), an NSF GRFP (DGE 1752814, https://www.nsfgrfp.org/) (EE, MT), a UC Berkeley Chancellor's Fellowship (EE, https://grad.berkeley.edu/admissions/apply/fellowships-entering/), a KFAS scholarship (YJK, https://www.kfas.or.kr/ScholarShip/ScholarShip0201.aspx?pCulture=en), and an 430 DoD NDSEG graduate fellowship (JL, https://ndseg.sysplus.com/). The funders had no role in study design, data collection and analysis, decision to publish, or preparation of the manuscript.

**Competing interests:** The authors have declared that no competing interests exist.

developed a computational framework for analyzing the transcription cycle that quantifies rates of RNA initiation, elongation, and cleavage, given input datasets from live cell imaging. Using the developing fruit fly embryo as a case study, we demonstrate that our methodology can quantitatively describe the whole transcription cycle at single-cell resolution. These results allow us to investigate a plethora of avenues, from couplings between different aspects of the transcription cycle at the single-cell level to comparisons with theoretical predictions of distributions of elongation rates across cells. We envision our methodology to provide a unified computational framework for the analysis of transcriptional data obtained from live cell imaging.

## Introduction

The eukaryotic transcription cycle consists of three main steps: initiation, elongation, and cleavage of the nascent RNA transcript (Fig 1A; [1]). Crucially, each of these three steps can be controlled to regulate transcriptional activity. For example, binding of transcription factors to enhancers dictates initiation rates [2], modulation of elongation rates helps determine splicing efficiency [3], and regulation of cleavage controls aspects of 3' processing such as alternative polyadenylation [4].

The steps of the transcription cycle can be coupled with each other. For example, elongation rates contribute to determining mRNA cleavage and RNA polymerase (RNAP) termination efficiency [5–8], and functional linkages have been demonstrated between transcription initiation and termination [9, 10]. Nonetheless, initiation, elongation, and transcript cleavage have largely been studied in isolation. In order to dissect the entire transcription cycle, it is necessary to develop a holistic approach that makes it possible to understand how the regulation of each step dictates mRNA production and to unearth potential couplings among these steps.

To date, the processes of the transcription cycle have mostly been studied in detail using *in vitro* approaches [11, 12] or genome-wide measurements that require the fixation of cellular material and lack the spatiotemporal resolution to uncover how the regulation of the transcription cycle unfolds in real time [13–18]. Only recently has it become possible to dissect these processes in living cells and in their full dynamical complexity using tools such as MS2 or PP7 to fluorescently label nascent transcripts at single-cell resolution [19–22]. These technological advances have yielded insights into, for example, intrinsic transcriptional noise in yeast [23], kinetic splicing effects in human cells [24], elongation rates in *Drosophila melanogaster* [25, 26], and transcriptional bursting in mammalian cells [27], *Dictyostelium* [28–30], fruit flies [25, 31–35] and *Caenorhabditis elegans* [36].

Despite the great promise of MS2 and PP7, using these techniques to comprehensively analyze the transcription cycle is hindered by the fact that the signal from these *in vivo* RNA-labeling technologies convolves contributions from all aspects of the cycle. Specifically, the fluorescence signal from nascent RNA transcripts persists throughout the entire cycle of transcript initiation, elongation, and cleavage; further, a single gene can carry many tens of transcripts. Thus, at any given point, an MS2 or PP7 signal reports on the contributions of transcripts in various stages of the transcription cycle [37]. Precisely interpreting an MS2 or PP7 signal therefore demands an integrated approach that accounts for this complexity.

Here, we present a method for analyzing live-imaging data from the MS2 and PP7 techniques in order to dynamically characterize the steps—initiation, elongation, and cleavage—of the full transcription cycle at single-cell resolution. While the transcription cycle is certainly more nuanced and can include additional effects such as sequence-dependent pausing [38], we

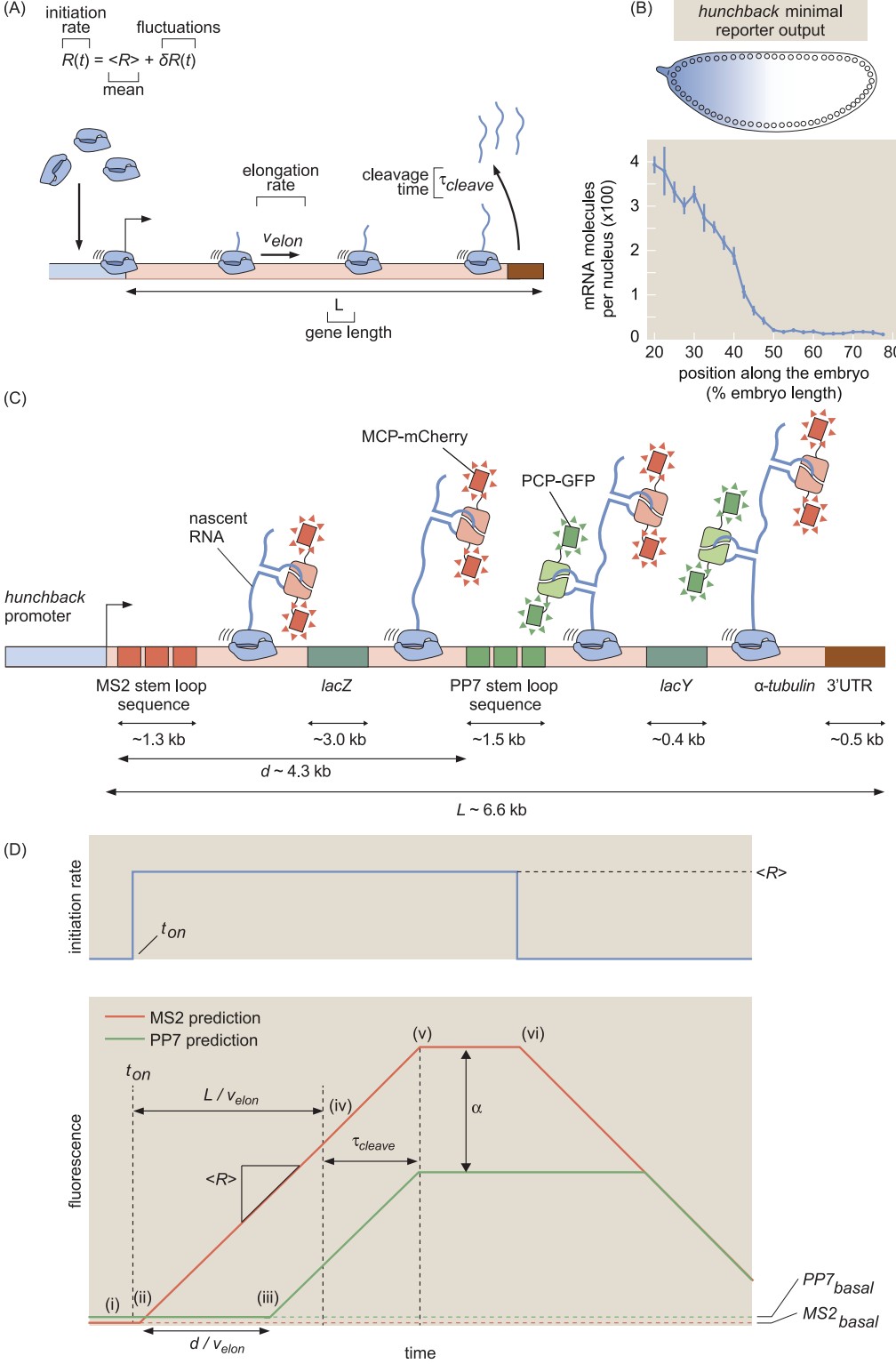

**Fig 1. Theoretical model of the transcription cycle and experimental setup.** (A) Simple model of the transcription cycle, incorporating nascent RNA initiation, elongation, and cleavage. (B) The reporter construct, which is driven by the *hunchback* P2 minimal enhancer and promoter, is expressed in a step-like fashion along the anterior-posterior axis of the fruit fly embryo. (C) Transcription of the stem loops results in fluorescent puncta with the 5' mCherry signal appearing before the signal from 3' GFP. Only one stem loop per fluorophore is shown for clarity, but the actual construct contains 24 repeats of each stem loop. (D, top) Relationship between fluorescence trace profiles and model parameters for an initiation rate consisting of a pulse of constant magnitude $\langle R \rangle$. (D, bottom, i) At first, the zero

initiation rate results in no fluorescence other than the basal levels $MS2_{basal}$ and $PP7_{basal}$ (red and green dashed lines). (ii) When initiation commences at time $t_{on}$, RNAP molecules load onto the promoter and elongation of nascent transcripts occurs, resulting in a constant increase in the MS2 signal (red curve). (iii) After time $\frac{d}{v_{elon}}$, the first RNAP molecules reach the PP7 stem loops and the PP7 signal also increases at a constant rate. (iv) After time $\frac{L}{v_{elon}}$, the first RNAP molecules reach the end of the gene, and (v) after the cleavage time $\tau_{cleave}$, these first nascent transcripts are cleaved. The subsequent loss of fluorescence is balanced by the addition of new nascent transcripts, resulting in a plateauing of the signal. (vi) Once the initiation rate shuts off, no new RNAP molecules are added and both fluorescence signals will start to decrease due to cleavage of the nascent transcripts still on the gene. Because elongation continues after initiation has ceased, the 5' MS2 signal begins decreasing before the 3' PP7 signal. The MS2 and PP7 fluorescent signals are rescaled to be in the same arbitrary units with the calibration factor $\alpha$. (Data in (B) adapted from [25] with the line representing the mean and error bars representing the standard error across 24 embryos).

view the quantification of these effective parameters as a key initial step for testing theoretical models. This method combines a dual-color MS2/PP7 fluorescent reporter [23, 24, 26] with Bayesian statistical inference techniques and quantitative modeling. As a proof of principle, we applied this analysis to the transcription cycle of a *hunchback* reporter gene in the developing embryo of the fruit fly *Drosophila melanogaster*. We validate our approach by comparing our inferred average initiation and elongation rates with previously reported results.

Crucially, our analysis also delivered novel single-cell statistics of the whole transcription cycle that were previously unmeasurable using genome-wide approaches, making it possible to generate distributions of parameter values necessary for investigations that go beyond simple population-averaged analyses [39–60]. We show that, by taking advantage of time-resolved data, our inference is able to filter out uncorrelated noise, such as that originating from random measurement error, in these distributions and retain sources of correlated variability (such as biological and systematic noise). By combining these statistics with theoretical models, we revealed substantial variability in RNAP stepping rates between individual molecules, demonstrating the utility of our approach for testing hypotheses of the molecular mechanisms underlying the transcription cycle and its regulation.

This unified analysis enabled us to investigate couplings between the various transcription cycle parameters at the single-cell level, whereby we discovered a surprising correlation of cleavage rates with nascent transcript densities. These discoveries illustrate the potential of our method to sharpen hypotheses of the molecular processes underlying the regulation of the transcription cycle and to provide a framework for testing those hypotheses.

## Results

To quantitatively dissect the transcription cycle in its entirety from live imaging data, we developed a simple model (Fig 1A) in which RNAP molecules are loaded at the promoter of a gene of total length $L$ with a time-dependent loading rate $R(t)$. For simplicity, we assume that each individual RNAP molecule behaves identically and independently: there are no interactions between molecules. While this assumption is a crude simplification, it nevertheless allows us to infer effective average transcription cycle parameters.

We parameterize this $R(t)$ as the sum of a constant term $\langle R \rangle$ that represents the mean, or time-averaged, rate of initiation, and a small temporal fluctuation term given by $\delta R(t)$ such that $R(t) = \langle R \rangle + \delta R(t)$. This mean-field parameterization is motivated by the fact that many genes are well approximated by constant rates of initiation [25, 31, 35, 61]. The fluctuation term $\delta R(t)$ allows for slight time-dependent deviations from the mean initiation rate. As a result, this term makes it possible to account for time-dependent behavior that can occur over the course of a cell cycle once the promoter has turned on. After initiation, each RNAP molecule traverses the gene at a constant, uniform elongation rate $v_{elon}$. Upon reaching the end of

the gene, there follows a deterministic cleavage time, $\tau_{cleave}$, after which the nascent transcript is cleaved.

We do not consider RNAP molecules that do not productively initiate transcription [62] or that are paused at the promoter [16], as they will provide no experimental readout. Based on experimental evidence [25], we assume that these RNAP molecules are processive, such that each molecule successfully completes transcription, with no loss of RNAP molecules before the end of the gene (see Section H in S1 File for a validation of this hypothesis).

## Dual-color reporter for dissecting the transcription cycle

As a case study, we investigated the transcription cycle of early embryos of the fruit fly *D. melanogaster*. Specifically, we focused on the P2 minimal enhancer and promoter of the *hunchback* gene during the 14th nuclear cycle of development; the gene is transcribed in a step-like pattern along the anterior-posterior axis of the embryo with a 26-fold modulation in overall mRNA count between the anterior and posterior ends (Fig 1B [25, 63–65]). As a result, the fly embryo provides a natural modulation in mRNA production rates, with the position along the anterior-posterior axis serving as a proxy for mRNA output.

To visualize the transcription cycle, we utilized the MS2 and PP7 systems for live imaging of nascent RNA production [25, 31, 33]. Using a two-color reporter construct similar to that reported in [23], [24], and [26], we placed the MS2 and PP7 stem loop sequences in the 5' and 3' ends, respectively, of a transgenic *hunchback* reporter gene (Fig 1C; see S1 Fig for more construct details). The *lacZ* sequence and a portion of the *lacY* sequence from *Escherichia coli* were placed as a neutral spacer [66] between the MS2 and PP7 stem loops.

As an individual RNAP molecule transcribes through a set of MS2/PP7 stem loops, constitutively expressed MCP-mCherry and PCP-GFP fusion proteins bind their respective stem loops, resulting in sites of nascent transcript formation that appear as fluorescent puncta under a laser-scanning confocal microscope (Fig 2A and S1 Video). The fluorescent signals did not exhibit noticeable photobleaching (Section B in S1 File and S2 Fig). Since *hunchback* becomes transcriptionally active at the start of the nuclear cycle before slowly decaying into a transcriptionally silent state [25, 67, 68], we restrict our analysis to the initial 18 minute window after mitosis where the promoter remains active.

The intensity of the puncta in each color channel is linearly related to the number of actively transcribing RNAP molecules that have elongated past the location of the associated stem loop sequence [25], albeit with different arbitrary fluorescence units. After reaching the end of the gene, which contains the 3'UTR of the $\alpha$-tubulin gene [66], the nascent RNA transcript undergoes cleavage. Because the characteristic timescale of mRNA diffusion is about two order of magnitudes faster than the time resolution of our experiment, we approximate the cleavage of a single transcript as resulting in the instantaneous loss of its associated fluorescent signal in both channels (Section C in S1 File; [69]). We included a few additional parameters in our model to make it compatible with this experimental data: a calibration factor $\alpha$ between mCherry and eGFP intensities, a time of transcription onset $t_{on}$ after mitosis at which the promoter switches on, and basal levels of fluorescence in each channel $MS2_{basal}$ and $PP7_{basal}$ (see Section A in S1 File for more details). The qualitative relationship between the model parameters and the fluorescence data is described in Fig 1D, which considers the case of a pulse of constant initiation rate.

## Transcription cycle parameter inference using Markov Chain Monte Carlo

We developed a statistical framework to estimate transcription-cycle parameters (Fig 1A) from fluorescence signals. Time traces of mCherry and eGFP fluorescence intensity are extracted

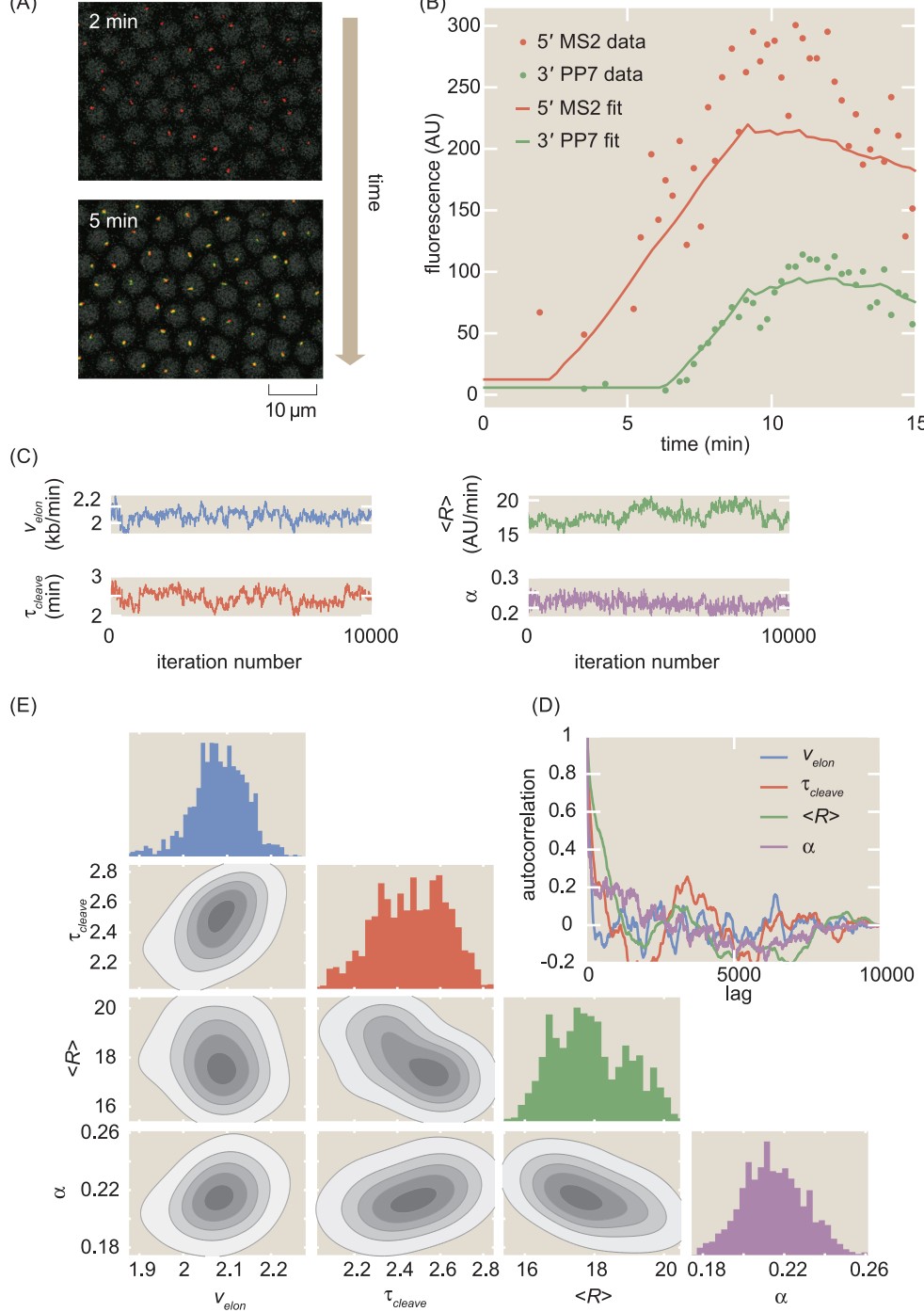

**Fig 2. MCMC inference procedure.** (A) Snapshots of confocal microscopy data over time, with MS2-mCherry (red) and PP7-eGFP (green) puncta reporting on transcription activity. Gray circles correspond to iRFP-labeled histones whose fluorescence is used as a fiduciary marker for cell nucleus segmentation (see Methods and materials for details). (B) Sample single-cell MS2 and PP7 fluorescence (points) together with best-fits of the model using MCMC inference (curves). (C) Raw MCMC inference chains for the elongation rate $v_{elon}$, cleavage time $\tau_{cleave}$, mean initiation rate $\langle R \rangle$, and calibration factor $\alpha$ for the inference results of a sample single cell. (D) Auto-correlation function for the raw chains in (A) as a function of lag (i.e. inference sample number). (E) Corner plot of the raw chains shown in (C).

from microscopy data such as shown in Fig 2A and S1 Video to produce a dual-signal readout of nascent RNA transcription at single-cell resolution (Fig 2B, data points; see Methods and materials for details). To extract quantitative insights from the observed fluorescence data, we used the established Bayesian inference technique of Markov Chain Monte Carlo (MCMC) [70] to infer the effective parameter values in our simple model of transcription: the calibration factor between mCherry and eGFP intensities $\alpha$, the time-dependent transcription initiation rate, separated into the constant term $\langle R \rangle$ and fluctuations $\delta R(t)$, the elongation rate $v_{elon}$, the cleavage time $\tau_{cleave}$, the time of transcription onset $t_{on}$, and the basal levels of fluorescence in each channel $MS2_{basal}$ and $PP7_{basal}$.

The details of the inference procedure are described in Section D in S1 File. Briefly, the inference was run separately for each single cell, yielding chains of sampled parameter values (Fig 2C). These resulting chains exhibited rapid mixing and rapidly decaying auto-correlation functions (Fig 2D), indicative of reliable fits. Corner plots of the fits indicated reasonable posterior distributions (Fig 2E).

From these single-cell fits, the mean value of each parameter's chain was retained for further analysis. The final dataset was produced by filtering with an automated procedure that relied on overall fit quality (Section F in S1 File and S4 Fig). This curation procedure did not introduce noticeable bias in the results (S4(G)–S4(I) Fig). A small minority of the rejected cells (S4(E) Fig) exhibited highly time-dependent behavior reminiscent of transcriptional bursting [71], which lies outside the scope of our model and is explored more in the Discussion. A sample fit is shown in Fig 2B. To aggregate the results, we constructed a distribution from the inferred parameter from each single-cell. Intra-embryo variability between single cells was greater than inter-embryo variability (Section I in S1 File and S6 Fig). As a result, unless stated otherwise, all statistics reported here were aggregated across 355 single cells combined between 7 embryos, and all shaded errors reflect the standard error of the mean.

## MCMC successfully infers calibration between eGFP and mCherry intensities

Due to the fact that the MS2 and PP7 stem loop sequences were associated with mCherry and eGFP fluorescent proteins, respectively, the two experimental fluorescent signals possessed different arbitrary fluorescent units, related by the scaling factor $\alpha$ given by

$$\alpha = \frac{F_{MS2}}{F_{PP7}}, \tag{1}$$

where $F_{MS2}$ and $F_{PP7}$ are the fluorescence values generated by a fully transcribed set of MS2 and PP7 stem loops, respectively. Although $\alpha$ has units of $AU_{MS2}/AU_{PP7}$, we will express $\alpha$ without units in the interest of clarity of notation.

We inferred single-cell values of $\alpha$ using the inference methodology. As shown in the blue histogram in Fig 3A, our inferred values of $\alpha$ possessed a mean of $0.145 \pm 0.004$ (SEM) and a standard deviation of 0.068.

As an independent validation, we measured $\alpha$ by using another two-color reporter, consisting of 24 alternating, rather than sequential, MS2 and PP7 loops [72–74] inserted at the 5' end of our reporter construct (Fig 3B). Thus, this reporter had a total of 48 stem loops, with 24 each of MS2 and PP7.

Fig 3C shows a representative trace of a single spot containing our calibration construct (see S2 Video for full movie). For each time point, the mCherry fluorescence in all measured single-cell traces was plotted against the corresponding eGFP fluorescence (Fig 3D, yellow points). The mean $\alpha$ was then calculated by fitting the resulting scatter plot to a line going

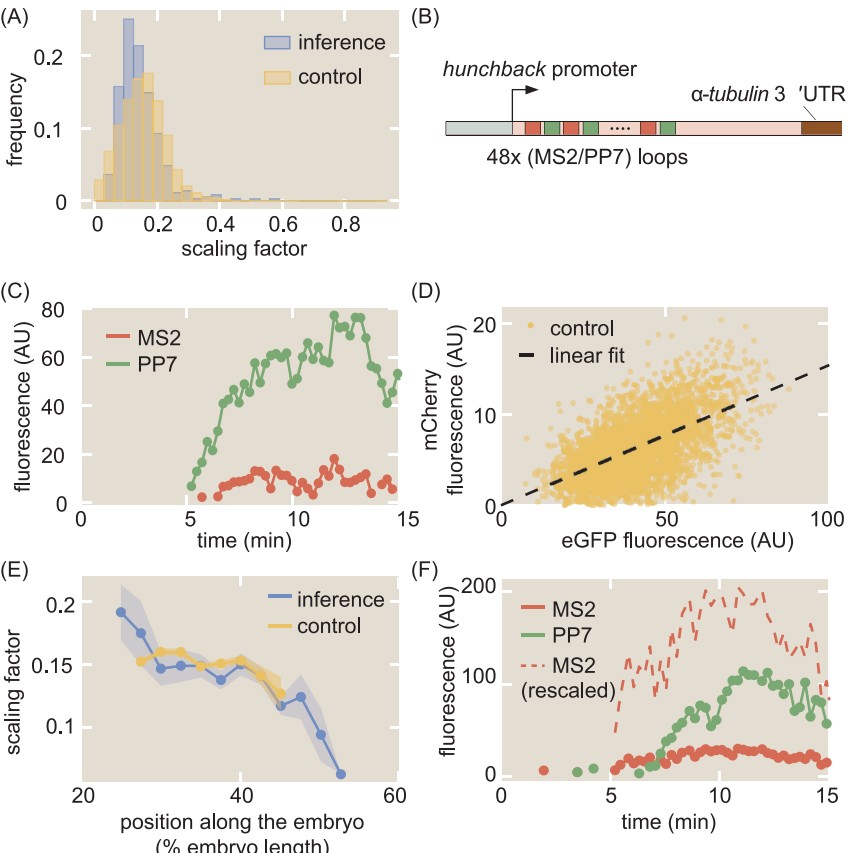

**Fig 3. Calibration of MS2 and PP7 fluorescence signals.** (A) Histogram of inferred values of $\alpha$ at the single-cell level from inference (blue), along with histogram of $\alpha$ values from the control experiment (yellow). (B) Schematic of construct used to measure the calibration factor $\alpha$ using 24 interlaced MS2/PP7 loops (48 loops in total). (C) Sample single-cell MS2 (red) and PP7 (green) traces from this control experiment. (D) Scatter plot of MS2 and PP7 fluorescence values for each time point (yellow) along with linear best fit (black) resulting in $\alpha = 0.154 \pm 0.001$. (E) Position-dependent mean value of $\alpha$ in both the inference (blue) and the control experiment (yellow). (F) Representative raw and rescaled MS2 and PP7 traces for a sample single cell in the inference data set. (A, D, E, data were collected for 314 cells across 4 embryos for the interlaced reporter, and for 355 cells across 7 embryos for the reporter with MS2 on the 5' and PP7 on the 3' of the gene (Fig 1C); shaded regions in (E) reflect standard error of the mean. Measurement conditions for both experiments are described in Methods and materials).

through the origin (Fig 3D, black line). The best-fit slope yielded the experimentally calculated value of $\alpha = 0.154 \pm 0.001$ (SEM). A distribution for $\alpha$ was also constructed by dividing the mCherry fluorescence by the corresponding eGFP fluorescence for each datapoint in Fig 3D, yielding the histogram in Fig 3A (yellow), which possessed a standard deviation of 0.073. Our independent calibration agreed with our inference, thus validating the infererred values of $\alpha$.

Interestingly, binning the cells by position along the embryo revealed a slight position dependence in the scaling factor. As shown in Fig 3E, both the directly measured and inferred $\alpha$ displayed higher values in the anterior, about 0.15, and lower values in the posterior, about 0.1. The fact that this position dependence is observed in both in the calibration experiments and inference suggests that this spatial modulation in the value of $\alpha$ is not an artifact of the constructs or our analysis, but a real feature of the system. We speculate that this spatial dependence could stem from differential availability of MCP-mCherry and PCP-GFP along the embryo, leading to a modulation in the maximum occupancy of the MS2 stem loops versus the PP7 stem loops [75].

Regardless, our data demonstrate that the inferred and calibrated $\alpha$ can be used inter-changeably, obviating the need for the control. Thus, the MS2 signals for each single cell could be rescaled to the same units as the PP7 signal (Fig 3F) within a single experiment, greatly increasing the power of the inference methodology. All plots, unless otherwise stated, reflect these rescaled values using the overall mean value of $\alpha = 0.145$ obtained from the inference.

## Inference of single-cell initiation rates recapitulates and improves on previous measurements

After validating the accuracy of our inference method in inferring transcription initiation, elongation, and cleavage dynamics using simulated data (Section G in S1 File and S5 Fig), we inferred these transcriptional parameters for the *hunchback* reporter gene as a function of the position along the anterior-posterior axis of the embryo. The suite of quantitative measurements on the transcription cycle produced by the aggregated inference results is shown in Fig 4A, 4C, 4E and 4F. Full distributions of these parameters can be found in S7 Fig.

Control of initiation rates is one of the predominant, and as a result most well-studied, strategies for gene regulation [2, 13, 76]. Thus, comparing our inferred initiation rates with previously established results comprised a crucial benchmark for our methodology. Our inferred values of the mean initiation rate $\langle R \rangle$ exhibited a step-like pattern along the anterior-posterior axis of the embryo, qualitatively reproducing the known *hunchback* expression profile (Fig 4A, blue). As a point of comparison, we also examined the mean initiation rate measured by [25], which was obtained by manually fitting a trapezoid (Fig 1D) to the average MS2 signal (Fig 4A, black). The quantitative agreement between these two dissimilar analysis methodologies demonstrates that our inference method can reliably extract the average rate of transcription initiation across cells.

Measurements of cell-to-cell variability in transcription initiation rate have uncovered, for example, the existence of transcriptional bursting and mechanisms underlying the establishment of precise developmental boundaries [39, 40, 45, 47, 48, 55, 57]. Yet, to date, these studies have mostly employed techniques such as single-molecule FISH to count the number of nascent transcripts on a gene or the number of cytoplasmic mRNA molecules [39–41, 43, 45, 48–52, 56–58, 77–84]. In principle, these techniques do not report on the variability in transcription initiation alone; they convolve this measurement with variability in other steps of the transcription cycle [76, 81].

Our inference approach isolates the transcription initiation rate from the remaining steps of the transcription cycle at the single-cell level, making it possible to calculate, for example, the coefficient of variation (CV; standard deviation divided by the mean) of the mean rate of initiation. Our results yielded values for the CV along the embryo that were fairly uniform, with a maximum value of around 40% (Fig 4F, blue). This value is roughly comparable to that obtained for *hunchback* using single-molecule FISH [45, 50, 57].

One of the challenges in measuring CV values, however, is that informative biological variability is often convolved with undesired experimental noise, such as experimental measurement noise inherent to fluorescence microscopy. In general, this experimental noise can contain both random, uncorrelated components as well as systematic components, the latter of which combines with actual biological variability to form overall correlated noise. Although we currently cannot entirely separate biological variability from experimental noise with our data and inference method, a strategy for at least separating uncorrelated noise from correlated noise was recently implemented in the context of snapshot-based fluorescent data [57]. By utilizing a dual-color measurement of the same biological signal, one can separate the total

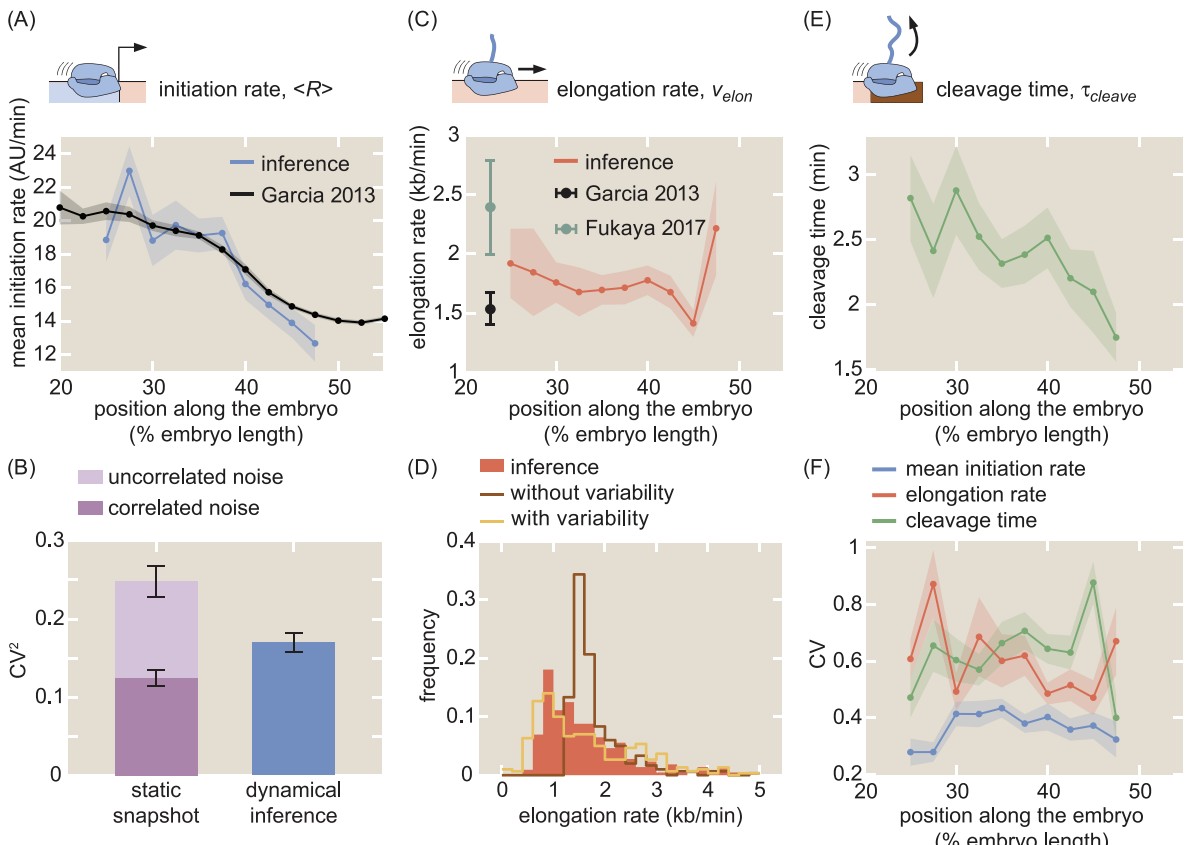

**Fig 4. Inferred transcription-cycle parameters.** (A) Mean inferred transcription initiation rate as a function of embryo position (blue), along with rescaled previously reported results (black, [25]). (B) Comparison of the squared CV of the mean initiation rate inferred using our approach (blue) or obtained from examining the fluorescence of transcription spots in a single snapshot (light plus dark purple). While snapshots captured a significant amount of uncorrelated noise (light purple), our inference accounts mostly for correlated noise (compare blue and dark purple). See Section K in S1 File and S8 Fig for details. (C) Inferred elongation rate as a function of embryo position (red), along with previously reported results (black, [25]; teal, [26]). (D) Distribution of inferred single-cell elongation rates in the anterior 40% of embryo (red), along with best fit to mean and standard deviation using single-molecule simulations with and without RNAP-to-RNAP variability (gold and brown, respectively, see Section M in S1 File for details). (E) Inferred cleavage time as a function of embryo position. (F) CV of the mean initiation rate (blue), elongation rate (red), and cleavage time (green) as a function of embryo position. (A, C, E, shaded error reflects standard error of the mean across 355 nuclei in 7 embryos, or of previously reported mean results; B, F, shaded error or black error bars represent bootstrapped standard errors of the CV or CV² for 100 bootstrap samples each; C, error bars reflect standard error of the mean for [25] and lower (25%) and upper (75%) quintiles of the full distribution from [26]).

variability in a dataset into uncorrelated measurement noise and correlated noise, which includes components such as true biological variability and systematic measurement error.

Building on this strategy, we first took a single snapshot from our live-imaging data and calculated the total squared CV of the fluorescence of spots at a single time point (Fig 4B, dark plus light purple). Compared to the squared CV from the inferred mean initiation rate (Fig 4B, blue), the squared CV from the snapshot was larger by about 0.1, suggesting that the inference method reported on a somewhat lower level of overall variability.

To investigate this disparity in measured variability further, we then rewrote the squared CV from the snapshot approach as the sum of uncorrelated and correlated noise components

$$CV^2_{total} = CV^2_{uncorrelated} + CV^2_{correlated}.$$ (2)

The magnitudes of each noise component were estimated by using the data from the interlaced reporter introduced in Fig 3B. To do so, we utilized the fact that, in principle, the mCherry and GFP signals from this experiment reflected the same underlying biological process, and assumed that deviations between the two signals were a result of uncorrelated measurement noise. Thus, we could apply the two-color formalism introduced in [85] to calculate the uncorrelated and correlated noise components from snapshots taken from the interlaced reporter construct (see Section K in S1 File and S8 Fig for more details).

The bar graph shown in Fig 4B shows that, once the uncorrelated noise (light purple) is subtracted from the total noise of our snapshot-based measurement, the remaining correlated variability (dark purple), which includes the biological variability, is slightly lower than the variability of our inference results (blue). Thus, our inference mostly captures correlated variability and filters out the bulk of the uncorrelated noise, similarly to techniques such as single-molecule FISH [57] but with the added advantage of also being able to resolve temporal information. Because such fixed tissue techniques ultimately provide static measurements that convolve signals from transcription initiation with those of elongation and cleavage, it is important to note that this is a qualitative comparison between the ability of fixed-tissue and live-imaging to separate correlated and uncorrelated variability. Thus, our results further validate our approach and demonstrate its capability to capture measures of cell-to-cell variability in the transcription cycle with high precision.

## Elongation rate inference reveals single-molecule variability in RNAP stepping rates

Next, we investigated the ability of our inference approach to report on the elongation rate $v_{elon}$. Nascent RNA elongation plays a prominent role in gene regulation, for example, in dosage compensation in *Drosophila* embryos [86], alternative splicing in human cells [3, 87], and gene expression in plants [88]. Our method inferred an elongation rate $v_{elon}$ that was relatively constant along the embryo (Fig 4C), lending support to previous reports indicating a lack of regulatory control of the elongation rate in the early fly embryo [26]. We measured a mean elongation rate of 1.72 ± 0.05 kb/min (SEM; n = 355), consistent with previous measurements of the fly embryo (Fig 4C, black and teal; [25, 26]), as well as with measurements from other techniques and model organisms, which range from about 1 kb/min to upwards of 4 kb/min [20, 23, 24, 27, 62, 76, 77, 89–92]. In addition, the CV of the elongation rate was roughly uniform across embryo position (Fig 4F, red).

Like cell-to-cell variability in transcription initiation, single-cell distributions of elongation rates can provide crucial insights into, for example, promoter-proximal pausing [54], traffic jams [93, 94], transcriptional bursting [95, 96], and noise propagation [59]. While genome-wide approaches have had huge success in measuring mean properties of elongation [16, 97], they remain unable to resolve single-cell distributions of elongation rates. We examined the statistics of single-cell elongation rates in the anterior 40% of the embryo, where the initiation rate was roughly constant, and inferred a broad distribution of elongation rates with a standard deviation of around 1 kb/min and a long tail extending to values upwards of 4 kb/min (Fig 4D, red). This large spread was consistent with observations of large cell-cell variability in elongation rates [76, 91] using a wide range of techniques, as well as with measurements from similar two-color live imaging experiments ([23, 26]; Section L in S1 File; S9 Fig).

To illustrate the resolving power of examining elongation rate distributions, we performed theoretical investigations of cell-to-cell variability in this transcription cycle parameter. Following [93], we considered a model where RNAP molecules stochastically step along a gene and cannot overlap or pass each other (Section M in S1 File). The model simulated MS2 and

PP7 fluorescences that were then run through the inference procedure, in order to account for the presence of inferential noise (Section G in S1 File).

First, we considered a scenario where the stepping rate of each RNAP molecule is identical. In this case, the sole driver of cell-to-cell variability is the combination of stochastic stepping behavior with traffic jamming due to steric hindrance of RNAP molecules. As shown in brown in Fig 4D, this model cannot account for the wide distribution of observed single-cell elongation rates.

In contrast, by allowing for substantial variability in the elongation rate of individual RNAP molecules, the model can reproduce the empirical distribution of single-cell elongation rates. As shown in gold in Fig 4D, the model can quantitatively approximate the inferred distribution within error (S9(D) Fig). This single-molecule variability is consistent with *in vitro* observations of substantial molecule-to-molecule variability in RNAP elongation rates [98, 99], thus demonstrating the ability of our approach to engage in the *in vivo* dissection of the transcription cycle at the single-molecule level.

## Inference reveals functional dependencies of cleavage times

Finally, we inferred values of the cleavage time $\tau_{cleave}$. Through processes such as alternative polyadenylation [4, 100] and promoter-terminator crosstalk [9, 10], events at the 3' end of a gene exert substantial influence over overall transcription levels [101]. Although many investigations of mRNA cleavage and RNAP termination have been carried out in fixed-tissue samples [102, 103], live-imaging studies with single-cell resolution of this important process remain sparse; some successes have been achieved in yeast and in mammalian cells [76]. We inferred a mean mRNA cleavage time in the range of 1.5–3 min (Fig 4E), consistent with values obtained from live imaging in yeast [22] and mammalian cells [24, 27, 62, 89]. Interestingly, as shown in Fig 4E, the inferred mRNA cleavage time was dependent on anterior-posterior positioning along the embryo, with high values ($\sim$3 min) in the anterior end and lower values toward the posterior end ($\sim$1.5 min). While the reasons for this position dependence are unknown, such dependence could result from the presence of a spatial gradient of a molecular species that regulates cleavage. Importantly, such a modulation could not have been easily revealed using genome-wide approaches that, by necessity, average information across multiple cells.

The CV of the cleavage time slightly increased toward the posterior end of the embryo (Fig 4F, green). Thus, although cleavage remains an understudied process compared to initiation and elongation, both theoretically and experimentally, these results provide the quantitative precision necessary to carry out such mechanistic analyses.

## Uncovering single-cell mechanistic correlations between transcription cycle parameters

In addition to revealing trends in average quantities of the transcription cycle along the length of the embryo, the simultaneous nature of the inference afforded us the unprecedented ability to investigate single-cell correlations between transcription-cycle parameters. We used the Spearman rank correlation coefficient ($\rho$) as a non-parametric measure of inter-parameter correlations. The mean initiation rate and the cleavage time exhibited a negative correlation ($\rho = -0.52$, $p$-val $\approx 0$; Fig 5A). This negative correlation at the single-cell level should be contrasted with the positive relation between these magnitudes at the position-averaged level, where the mean initiation rate and cleavage time both increased in the anterior of the embryo (Fig 4A and 4E). Thus, our analysis unearthed a quantitative relationship that was obscured by a naive investigation of spatially averaged quantities, an approach often used in fixed [57] and live-

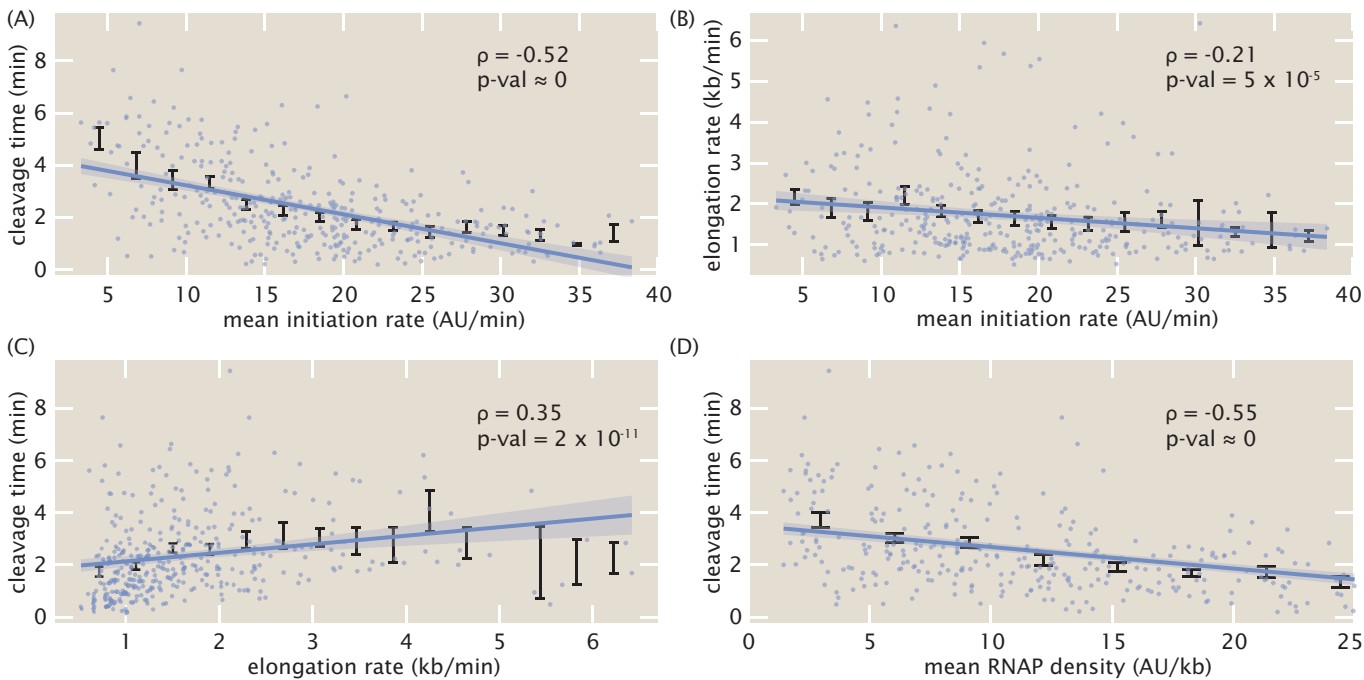

**Fig 5. Single-cell correlations between transcription cycle parameters.** Spearman rank correlation coefficients and associated p-values between (A) mean initiation rate and cleavage time, (B) mean initiation rate and elongation rate, (C) elongation rate and cleavage time, and (D) mean RNAP density and cleavage time. Blue points indicate single-cell values; black points and error bars indicate mean and SEM, respectively, binned across x-axis values. Lines and shaded regions indicate generalized linear model fit and 95% confidence interval, respectively, and are shown for ease of visualization (see Methods and materials for details).

imaging [35] studies, as well as in genome-wide investigations [104, 105]. We also detected a small negative correlation ($\rho = -0.21$, $p$-val = $5 \times 10^{-5}$) between elongation rates and mean initiation rates (Fig 5B). Finally, we detected a small positive correlation ($\rho = 0.35$, $p$-val = $2 \times 10^{-11}$) between cleavage times and elongation rates (Fig 5C). These results are consistent with prior studies implicating elongation rates in 3' processes such as splicing and alternative polyadenylation: slower elongation rates increased cleavage efficiency [3, 5].

The observed negative correlation between cleavage time and mean initiation rate (Fig 5A), in conjunction with the positive correlation between cleavage time and elongation rate (Fig 5C), suggested a potential underlying biophysical control parameter: the mean nascent transcript density on the reporter gene body $\rho$, given by

$$\rho = \frac{\langle R \rangle}{v_{elon}}. \tag{3}$$

Possessing units of (AU/kb), this mean transcript density estimates the average number of nascent RNA transcripts per kilobase of template DNA. Plotting the cleavage time as a function of the mean transcript density yielded a negative correlation ($\rho = -0.55$, $p$-val $\approx 0$) that was stronger than any of the other correlations between transcription-cycle parameters at the single-cell level (Fig 5D). Mechanistically, the correlation between cleavage time and mean transcript density suggests that, on average, more closely packed nascent transcripts at the 3' end of a gene cleave faster.

Further investigations using simulations indicated that this relationship did not arise from spurious correlations in the inference procedure itself (Section G in S1 File and S5(E)–S5(H) Fig),

but rather captured real correlations in the data. Furthermore, although the four inter-parameter correlations investigated here only used mean values obtained from the inference methodology, a Monte Carlo simulation involving the full Bayesian posterior distribution confirmed the significance of the results (Section N in S1 File and S11 Fig).

Using an absolute calibration for a similar reporter gene [25] led to a rough scaling of 1 AU ≈ 1 molecule corresponding to a maximal RNAP density of about 20 RNAP molecules/kb in Fig 5D. With a DNA footprint of 40 bases per molecule [106], this calculation suggests that, in this regime, RNAP molecules are densely distributed, occupying about 80% of the reporter gene. We hypothesize that increased RNAP density could lead to increased pausing as a result of traffic jams [93, 94]. Due to this pausing, transcripts would be more available for cleavage, increasing overall cleavage efficiency. Regardless of the particular molecular mechanisms underlying our observations, we anticipate that this ability to resolve single-cell correlations between transcription parameters, combined with perturbative experiments, will provide ample future opportunities for studying the underlying biophysical mechanisms linking transcription processes.

## Discussion

Over the last two decades, the genetically encoded MS2 [19] and PP7 [21] RNA labeling technologies have made it possible to measure nascent and cytoplasmic RNA dynamics *in vivo* in many contexts [20, 22–28, 31–36, 61, 62, 73, 76, 107–110]. However, such promising experimental techniques can only be as powerful as their underlying data-analysis infrastructure. For example, while initial studies using MS2 set the technological foundation for revealing transcriptional bursts in bacteria [20], single-celled eukaryotes [28, 111], and animals [25, 31], only recently did analysis techniques become available to reliably obtain parameters such as transcriptional burst frequency, duration, and amplitude [24, 35, 112–114].

In this work, we established a novel method for inferring quantitative parameters of the entire transcription cycle—initiation, elongation and cleavage—from live imaging data of nascent RNA dynamics. Notably, this method offers high spatiotemporal resolution at the single-cell level, resolving aspects of transcriptional activity within the body of an organism and at sub-minute resolution. Furthermore, while our experimental setup utilized two fluorophores, we found that the calibration between their intensities could be inferred directly from the data (Fig 3), rendering independent calibration and control experiments unnecessary.

After validating previously discovered spatial modulations in the mean initiation rate, we discovered an unreported modulation of the cleavage time with respect to embryo position that mirrored that of the mean initiation rate (Fig 4E). Although such a relationship at first would suggest a positive correlation between initiation and cleavage, the presence of significant negative correlation at the single-cell level refutes this idea (Fig 5A). Instead, we speculate that the spatial modulation of the cleavage time could underlie a coupling with a spatial gradient of some molecular factor that controls this transcription cycle parameter [115], possibly due to effects such as gene looping [116, 117].

These features are unattainable by widespread, but still powerful, genome-wide techniques that examine fixed samples, such as global run-on sequencing (GRO-seq) to measure elongation rates *in vivo* [118, 119]. Additionally, while fixed-tissue technologies such as single-molecule RNA-FISH provide superior spatial and molecular resolution to current live imaging technologies [45, 57], the fixation process necessarily prevents temporal analysis of the same single cell to study these dynamic transcriptional processes. Thus, live imaging approaches offer a complementary approach to widespread RNA-FISH studies of transcriptional dynamics [39–41, 43, 45, 48–52, 56–58, 77–84].

## Dissecting the transcription cycle at the single-cell level

From elucidating the nature of mutations [120] and revealing mechanisms of transcription initiation [23, 40, 42, 43, 45–48, 50, 57, 60, 95, 96], transcription elongation [54, 59, 121], and translational control [122], to enabling the calibration of fluorescent proteins in absolute units [123–128], examining single-cell distributions through the lens of theoretical models has made it possible to extract molecular insights about biological function that are inaccessible through the examination of averaged quantities. The single-cell measurements afforded by our approach made it possible to infer full distributions of transcription parameters (Fig 4B, 4D and 4F). This single-cell resolution motivates a dialogue between theory and experiment for studying transcription initiation, elongation, and cleavage at the single-cell level.

We showed how our inferred distributions of initiation rates effectively filter out most uncorrelated measurement noise, which we expect to be dominated by experimental noise, while retaining information on sources of correlated noise, including underlying biological variability (Fig 4B). Additionally, our theoretical model of elongation rate distributions make it possible to test mechanistic models of RNAP transit along the gene. While still preliminary and far from conclusive, our results suggest that cell-to-cell variability in elongation rates arises from single-molecule variability in stepping rates, and that processes such as stochasticity in stepping behavior and traffic jamming due to steric hindrance alone cannot account for the observed elongation rate distributions (Fig 4D). Such statistics could then be harnessed to make predictions for future perturbative experiments that utilize, for example, mutated RNAP molecules with altered elongation rates [129] or reporter genes with differing spacer lengths between MS2 and PP7 stem loops sequences.

Finally, the simultaneous single-cell inference of transcription-cycle parameters granted us the novel capability to investigate couplings between transcription initiation, elongation, and cleavage, paving the way for future studies of mechanistic linkages between these processes. In particular, the observed coupling of the mRNA cleavage time with RNAP density (Fig 5D) suggests future experiments utilizing, for example, orthogonal stem loops on either side of the 3'UTR as potential avenues for investigating mechanisms such as RNAP traffic jams [93, 94], inefficient or rate-limiting nascent RNA cleavage [7, 100], and promoter-terminator looping [130]. Other potential experiments could include perturbative effects, such as introducing inhibitors of transcription initiation, elongation, and/or cleavage and assessing the downstream impact on the inferred transcriptional parameters to see if the perturbed effects are separable or convolved between parameters.

## Comparison to existing analysis techniques

Our method provides a much-needed framework for applying statistical inference for the analysis of live imaging data of nascent transcription, complementing existing Bayesian approaches [131, 132] as well as expanding the existing repertoire of model-driven statistical techniques to analyze single-cell protein reporter data [133–136]. In particular, compared to auto-correlation analysis of transcriptional signals [137], another powerful method of analyzing live imaging transcription data, our method is quite complementary.

First, auto-correlation analysis typically requires a time-homogeneous transcript initiation process [137], and benefits immensely from having experimental data acquired over long time windows to enhance the auto-correlation signal (although recent work has improved on the ability to analyze short time windows [112]). In contrast, our model-driven inference approach can account for slight time dependence and can fit short time traces. This is of particular

relevance to the fly embryo, where each cell cycle in early development is incredibly short (here, we only examined 18 minutes of data) and transcription initiation switches from OFF to ON and back to OFF within that timeframe.

Second, auto-correlation analysis depends strongly on signal-noise ratio, namely the ability to resolve single-or-few-transcript fluctuations in the number of actively transcribing polymerases on a gene [22, 24]. Our approach, however, can be applied even if the signal-noise ratio can only resolve differences in transcript number of several transcripts, rather than just one.

Third, our model-driven approach benefits from explicitly parameterizing the various steps of the transcription cycle, allowing for the separation of processes such as elongation and cleavage. In contrast, while the auto-correlation technique has the advantage of not relying on a particular specific model, it does rely on unknown parameters such as the overall transcript dwell time, which is a combination of elongation and cleavage. Thus, it becomes harder to separate contributions from these different processes. Additionally, auto-correlation approaches cannot produce absolute rates of transcriptional processes, such as the quantified rates of mean transcription initiation obtained in this work.

## Future improvements

Future improvements to experimental or inferential resolution could sharpen precision of single-cell results, increasing confidence in the distributions obtained through this methodology. For example, technologies such as lattice light-sheet microscopy [138–142] would vastly improve spatiotemporal imaging resolution and reduce uncertainty in measurements. While this increased resolution is unlikely to dramatically change the statistics reported here, it could potentially push the analysis regime to the single-molecule level, necessitating the parallel development of increasingly refined models that can account for stochasticity and fluctuations that are not resolved with bulk measurements. In addition, while our analysis restricted itself to consider only nascent RNA labeling technologies, this methodology could be extended to also examine mature labeled RNA in the nucleus and cytoplasm of an organism, providing a more complete picture of transcription.

One important caveat of our method is the failure to account for genes that undergo transcriptional bursting [71]. Here, the initiation rate fluctuates much more rapidly in time such that our assumption of a constant mean transcription initiation rate breaks down. We chose not to address this regime in this work because only a small minority of cells (4%) studied exhibited bursting behavior. Nevertheless, although our model does not capture bursting behavior (Section F in S1 File; S4(E) and S4(F) Fig), transcriptional bursting remains a prevalent phenomenon in eukaryotic transcription and thus motivates extensions to this work to account for its behavior. For example, one possible implementation to account for transcriptional bursting could first utilize the widespread two-state model used to describe this phenomenon [141] in order to partition a time trace into ON and OFF time windows. Then the MCMC inference method developed in this work could be used to quantify the transcription cycle during the ON and OFF windows with finer precision.

## Outlook

To conclude, while we demonstrated this inference approach in the context of the regulation of a *hunchback* reporter in *Drosophila melanogaster*, it can be readily applied to other genes and organisms in which MS2 and PP7 have been already implemented [20, 25, 27, 28, 31, 36, 62, 142], or where non-genetically encoded RNA aptamer technologies such as Spinach [142, 143] are available. Thus, we envision that our analysis strategy will be of broad applicability to

the quantitative and molecular *in vivo* dissection of the transcription cycle and its regulation across many distinct model systems.

## Methods and materials

### DNA constructs

The fly strain used to express constitutive MCP-mCherry and PCP-eGFP consisted of two transgenic constructs. The first construct, MCP-NoNLS-mCherry, was created by replacing the eGFP in MCP-NoNLS-eGFP [25] with mCherry. The second construct, PCP-NoNLS-eGFP, was created by replacing MCP in the aforementioned MCP-NoNLS-eGFP with PCP, sourced from [22]. Both constructs were driven with the *nanos* promoter to deliver protein maternally into the embryo. The constructs lacked nuclear localization sequences because the presence of these sequences created spurious fluorescence puncta in the nucleus that decreased the overall signal quality. Both constructs were incorporated into fly lines using P-element transgenesis, and a single stable fly line was created by combining all three transgenes.

The reporter construct P2P-MS2-lacZ-PP7 was cloned using services from GenScript. It was incorporated into the fly genome using PhiC31-mediated Recombinase Mediated Cassette Exchange (RMCE) [144], at the 38F1 landing site.

Full details of construct and sequence information can be found in a public Benchling folder.

### Fly strains

Transcription of the *hunchback* reporter was measured by imaging embryos resulting from crossing *yw;MCP-NoNLS-mCherry, Histone-iRFP;MCP-NoNLS-mCherry, PCP-NoNLS-GFP* female virgins with *yw;P2P-MS2-LacZ-PP7* males. The *Histone-iRFP* transgene was provided as a courtesy from Kenneth Irvine and Yuanwang Pan.

### Sample preparation and data collection

Sample preparation followed procedures described in [32], [145], and [35]. To summarize, embryos were collected, dechorinated with bleach and mounted between a semipermeable membrane (Lumox film, Starstedt, Germany) and a coverslip while embedded in Halocarbon 27 oil (Sigma). Excess oil was removed with absorbent paper from the sides to flatten the embryos slightly. Data collection was performed using a Leica SP8 scanning confocal microscope (Leica Microsystems, Biberach, Germany). The MCP-mCherry, PCP-eGFP, and Histone-iRFP were excited with laser wavelengths of 488 nm, 587 nm, and 670 nm, respectively, using a White Light Laser. Average laser powers on the specimen (measured at the output of a 10x objective) were 35 $\mu$W and 20 $\mu$W for the eGFP and mCherry excitation lasers, respectively. Three Hybrid Detectors (HyD) were used to acquire the fluorescent signal, with spectral windows of 496–546 nm, 600–660 nm, and 700–800 nm for the eGFP, mCherry, and iRFP signals, respectively. The confocal stack consisted of 15 equidistant slices with an overall z-height of 7 $\mu$m and an inter-slice distance of 0.5 $\mu$m. The images were acquired at a time resolution of 15 s, using an image resolution of 512 x 128 pixels, a pixel size of 202 nm, and a pixel dwell time of 1.2 $\mu$s. The signal from each frame was accumulated over 3 repetitions. Data were taken for 355 cells over a total of 7 embryos, and each embryo was imaged over the first 25 min of nuclear cycle 14.

## Image analysis

Images were analyzed using custom-written software following the protocols in [25] and [35]. This software contains MATLAB code automating the analysis of all microscope images obtained in this work, and can be found on a public GitHub repository. Briefly, this procedure involved segmenting individual nuclei using the Histone-iRFP signal as a nuclear mask, segmenting each transcription spot based on its fluorescence, and calculating the intensity of each MCP-mCherry and PCP-eGFP transcription spot inside a nucleus as a function of time. The Trainable Weka Segmentation plugin for FIJI [146], which uses the FastRandomForest algorithm, was used to identify and segment the transcription spots. The final intensity of each spot over time was obtained by integrating pixel intensity values in a small window around the spot and subtracting the background fluorescence measured outside of the active transcriptional locus. When no activity was detected, a value of NaN was assigned.

## Data analysis

Inference was done using *MCMCstat*, an adaptive MCMC algorithm [147, 148]. Figures were generated using the open-source gramm package for MATLAB, developed by Pierre Morel [149]. Generalized linear regression used in Fig 5 utilized a normally distributed error model and was performed using MATLAB's *glmfit* function. All scripts relating to the MCMC inference method developed in this work are available at the associated GitHub repository.

## Supporting information

**S1 File. Detailed information supporting main text. Table A:** Mean and standard deviation of model parameters used in single-cell simulations. **Table B:** Comparison of Spearman rank correlation coefficients and p-values between experimental and simulated single-cell correlations. **Table C:** Parameters used in single-molecule Monte Carlo simulation of elongation rates.
(PDF)

**S1 Fig. Detailed description of reporter construct used in this work.**
(EPS)

**S2 Fig. Investigation of photobleaching in experimental setup.**
(EPS)

**S3 Fig. Scaling of fluorescence measurement noise with overall fluorescence intensity.**
(EPS)

**S4 Fig. Automated curation of data.**
(EPS)

**S5 Fig. Overview of MCMC inference validation.**
(EPS)

**S6 Fig. Comparison of intra- and inter-embryo variability for inferred mean initiation rates, elongation rates, and cleavage times, as a function of embryo position.**
(EPS)

**S7 Fig. Single cell distributions of inferred parameters.**
(EPS)

**S8 Fig. Comparison of coefficients of variation (CV) between inferred mean initiation rates and instantaneous counts of number of nascent RNA transcripts.**
(EPS)

**S9 Fig. Comparison of distribution of elongation rates with previous studies.**
(EPS)

**S10 Fig. Single-molecule simulations of elongation dynamics require molecular variability to describe empirical distributions.**
(EPS)

**S11 Fig. Monte Carlo simulation of error in single-cell analysis.**
(EPS)

**S1 Video. Measurement of main reporter construct.**
(AVI)

**S2 Video. Measurement of interlaced reporter construct.**
(AVI)

**S1 Data. Dataset containing results from inference procedure and simulations.**
(ZIP)

## Acknowledgments

We thank Sandeep Choubey, Antoine Coulon, Jane Kondev, Anders Sejr Hansen, Mustafa Mir, Rob Phillips, Manuel Razo-Mejia, and Matthew Ronshaugen for thoughtful comments on the manuscript. We also are grateful to Florian Jug, Nick Lammers, and Armando Reimer for their crucial work in developing the image analysis code used here.

## Author Contributions

**Conceptualization:** Jonathan Liu, Hernan G. Garcia.

**Data curation:** Jonathan Liu, Donald Hansen.

**Formal analysis:** Jonathan Liu.

**Funding acquisition:** Hernan G. Garcia.

**Investigation:** Jonathan Liu, Donald Hansen.

**Methodology:** Jonathan Liu, Donald Hansen.

**Project administration:** Jonathan Liu, Hernan G. Garcia.

**Resources:** Jonathan Liu, Elizabeth Eck, Yang Joon Kim, Meghan Turner, Simon Alamos.

**Software:** Jonathan Liu.

**Supervision:** Jonathan Liu.

**Validation:** Jonathan Liu.

**Visualization:** Jonathan Liu.

**Writing – original draft:** Jonathan Liu.

**Writing – review & editing:** Jonathan Liu, Hernan G. Garcia.

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
