## [Decision Letter · Decision Letter 0]

22 Dec 2020

Dear Mr. Liu,

Thank you very much for submitting your manuscript "Single-cell characterization of the eukaryotic transcription cycle using live imaging and statistical inference" for consideration at PLOS Computational Biology.

As with all papers reviewed by the journal, your manuscript was reviewed by members of the editorial board and by several independent reviewers. In light of the reviews (below this email), we would like to invite the resubmission of a significantly-revised version that thoroughly addresses the reviewers' concerns.

We cannot make any decision about publication until we have seen the revised manuscript and your response to the reviewers' comments. Your revised manuscript is also likely to be sent to reviewers for further evaluation.

Sincerely,

James R. Faeder

Associate Editor

PLOS Computational Biology

Jian Ma

Deputy Editor

PLOS Computational Biology

Reviewer's Responses to Questions

**Comments to the Authors:**

Reviewer #1: Regulation of transcription is a fundamental problem that has received growing attention with the emergence of technologies that enable single-cell measurements. In their manuscript, Liu et al hybridize experimental and computational approaches to investigate single cell variability of biophysical parameters for the transcription cycle across the AP axis of developing fruit fly embryos. Nascent mRNA production is quantified over time in living cells using orthogonal fluorescence-labeled stem loops placed at the 3’ and 5’ ends of a transgenic reporter gene. Biophysical parameters are subsequently calculated for each cell using Bayesian inference and mechanistic modeling. The resultant fits reveal significant cell-to-cell variability in rates for transcriptional initiation, elongation, and cleavage, and also support a range for intermolecular variability between individual RNA polymerases. Value in the approach for hypothesis generation/refinement is furthermore demonstrated with parameter co-variation analysis, suggesting novel mechanistic relationships between steps of the transcription cycle that are likely to be the topic of follow-up studies. Although the manuscript can still be improved by addressing several issues, it is elegant in its simplicity and of high technical quality. Overall, I expect the optimized approach and results will valuable to the readership of PLoS Computational Biology and should be published following revision.

-RECL

Major Comments:

1. My predominant concern relates to interpretation of the data. In supplementary movie 1, it’s clear that mCherry peaks in intensity and fades before the eGFP channel approaches its peak in intensity in the same cell (further supported by the data points for a single cell in figure 2b). This seems in contrast with the assumptions of RNA processivity and instantaneous cleavage in timescales of the experiment. Based on these assumptions my expectation is that decay in intensity for both fluorescent molecules should begin and proceed simultaneously. Is there an interpretation of this phenomenon that does not conflict with the assumptions?

2. During curation, roughly half of the single cell data that passed technical requirements are filtered and discarded, leaving a subset of trajectories that can be well-fit by the model. It should be reported what fraction of filtered cells are low signal:noise (S3E) and the fraction trajectories with poor fits (S3F). If the poor fits are significant in proportion, does this class of trajectories have any common features? An unfortunate requirement of standard Bayesian inference is that a model topology is itself assumed true. In combination with comment 1, it seems likely that the model, or its underlying assumptions, may be inadequate to reflect true biological complexity (which is probably always the case), but this has several consequences – one of which is that biophysical inferences may reflect a combination of biological rates associated with multiple cellular processes. The possibility exists that cells filtered because of poor fit indicate other states of the system that are not recapitulated in the model topology, but are reflected in another topology. Some of these limitations and ramifications associated with Bayesian inference would make relevant discussion at the discretion of the authors/editor.

3. The authors perform a nice control experiment treating their movies as snapshot data to infer sources of correlated and uncorrelated noise. Because CVs of their inference approach are comparable with the biological noise component from the snapshot analysis, the authors conclude that inference filters experimental noise and effectively captures biological variability only. It’s not fully clear what criteria are used to establish that inference is capturing only biological variability only in Figs 2D and S7, and not some combination of variability with noise sources? Although it may be true that inference is filtering poisson noise, I can imagine other systematic sources of noise that would almost certainly be reflected in inferred parameters that do not reflect variation in a transcriptional parameters. For example, compare parameters inferred for a trajectory and for the same trajectory convolved with the line y = at+b which may represent a time varying fluctuation in the fluorescent coat protein – this type of noise will not be filtered by Bayesian inference. In addition, for this assertion to be true, the snapshot analyses must be limited to only the same curated cells that were analysed by inference (i.e. the ~30% of cells that passed the stringent filters), although it’s not clear that this is the case.

4. Although the writing and presentation is generally very clear, the results section seems awfully long and wordy, and requires regular flipping between the MS and supplement. For example, much of Figure 2 establishes that rates from inference are consistent with previous observations, yet the description is spread over 6 pages of text that require significant detail from various subsections of the supplement. A constructive criticism is to distill the results section, moving text as necessary to the intro/discussion, and recover some of the details from the supplement into the main paper. In my opinion, Figure S2 is an important representation of the fundamental approach that would be appreciated as a main figure by the computational readership, and there may be others at the discretion of the authors/editor.

Minor Comments:

1. The abstract builds the expectation that there is significant novelty in the computational algorithm, but the core approach is standard Bayesian inference with a few optimizations for these particular data. At the authors’ discretion, I’d suggest edits to the abstract and introduction as necessary to keep these expectations realistic.

2. S3.3 curation suggests 260 cells were skipped from 1053 total leaving 567: these numbers don’t add up.

3. Explanation/citation for the ranges of prior distributions in the supplement

4. Full model seems to assume instantaneous and complete binding of fluorescent protein-fused PCP/MCP to stem loops. The assumption should be explicit and possible caveats discussed in the relevant section of the supplement.

5. More details are necessary to understand why simulated trajectories have poor S:N (line 931).

6. The discussion could benefit from description of how the experimental limitations (15s/frame) and inference limitations may be improved in the future. Also whether inferred parameters are likely to change significantly with more technological advancements.

Reviewer #2: The authors combine quantitative experiments of nascent transcription with computational modeling to simultaneously study the relationship between critical steps in transcription: initiation, elongation, and cleavage on nascent RNA. Studying these three processes together rather than independently, as often done in the literature, is an essential step toward understanding the transcription cycle. Integration of experiments with computational modeling is vital in this process.

I am overall enthusiastic about the problem and the importance of integration of modeling with experiments. I am also excited about the fact that the experiments have been performed in a multicellular organism. However, I have major reservations about why a model is necessary in the first place? What is the utility of the model besides data analysis? I also have concerns about the implementation and choice of modeling tools. Besides, I have concerns about the lack of cited references related to the same and competing modeling approaches from other groups. Moreover, I have reservations about the presentation of the results. Lastly, I have concerns about the lack of controls.

Here are my specific concerns:

1. The model is not used to make any predictions for new or non analyzed data. Although I see the value in using a model to quantify the single-cell data, I have concerns about how predictive such a model is. How do the results change if one would use a piecewise linear fit to each of the sections of the single cell transcription time trace?

2. A discussion should be included why there are differences in the distribution shape between the inferred single-cell parameters from the model fit and the distributions from the simulations (e.g. Fig2F). I believe the simulations make assumptions about the shape of the parameter distribution that differs from the data.

3. An explanation should be provided why R(t) is parameterized as a constant and a noise term. What are the advantages and disadvantages of this approach? The explanation should also include modeling approaches from other groups to understand the context of the work better.

4. The authors stress in their abstract that the in-vivo dissection of initiation, elongation, and cleavage is challenging because of the lack of sufficient spatiotemporal resolution to separate the contributions from each of these steps. It is not clear in the context of the manuscript what the authors mean by that. Are you referring to the lack of spatiotemporal analysis of nascent, nuclear, and cytoplasmic RNA or referring to differences in expression within the organism? Both are important questions that could be answered in this model system. A differentiated discussion would be helpful for the manuscript, particularly in the context of the findings.

5. Also, in the abstract, the authors claim that the cell-to-cell variability is due to variability in the inferred parameters. It is not clear to me that this kind of statement is the only explanation. Alternative possibilities should be discussed that could contribute to these results.

6. A discussion should also include the advantages and disadvantages of measurement and model inference from live cells compared to snapshots in time of single-cell measurements of transcription (RNA-FISH).

7. The presentation of the data needs to improve substantially. Currently, many of the critical results are buried in the supplementary material. It is also unclear the intermediate steps from the concept to the final results as shown in Fig.2.

8. The author state that they measured 299 cells across 7 embryos. In the title and abstract, the authors state that they study cell to cell variability. So why are many of the plots showing population-averaged data? It would be much more insightful if the results are plotted as joint probability distributions for any of the graphs, instead of showing population averages, to appreciate the level of variability truly.

9. More of the supplementary figures should be moved to the main manuscript. I don’t see any reason why the main manuscript needs to have only 3 figures.

10. Interestingly, some of the parameters change within the embryo (lines 260-262). Unfortunately, the biological reason for this is not discussed, nor how the organism could regulate these changes. More discussion should be added to better present this exciting result.

11. Figure 1 outlines the different steps in the transcription cycle that can be inferred from the single-cell time-lapse data. Figure S1 should be integrated into Figure1. All the features that can be extracted from the single-cell time traces should also be included in this figure to make readers fully aware of the approach's power.

12. Besides the authors filtering out cells that do not have enough time points measured, the authors filter out 268 cells (567-299 retained after curation) that could not be inferred. This might bias their results if they filter data that their model can describe vs. improve/alter their model to describe those cells. The concern is that filtering might be removing some dynamics that could not be inferred by the model instead of just quality control. Experimental acquisition noise/Intrinsic Biological deviation from the model does not seem to me be enough justification for removing cells from the dataset. A comparison between the current data and the data with this additional curve should be compared to remove this does not change the results.

13. No experiments are performed testing if rates can be separated from each transcription cycle step. Potential further experiments would be to add inhibitors of regulatory proteins involved in nascent transcription initiation, elongation, and or cleavage to see if specific rates in the model associated with each transcription cycle change.

14. How does the insertion of the MS2 / PP7 repeat impact the transcription cycle? Comparing live-cell labeled nascent transcription with RNA-FISH on fixed cells with probes against MS2 or PP7 repeat is required to ensure that the repleads do not introduce artifacts.

Minor:

15. The statistical analysis in figure three was not the best suited for the data. The R^2 value is essentially meaningless due to the data not fitting a line, and linear regression, in general, is not ideal for data that is not normally distributed. I would suggest using a non-parametric test for correlation, such as the Spearman.

16. Inline 197, the authors state: “...convolved with undesired experimental noise.”. What does this mean? Please provide more detail.

17. The manuscript's title is a summary of what has been done. But not the take-home message of the manuscript? I recommend rephrasing the title.

Reviewer #3: the review uploaded as an attachment

Reviewer #4: %%%% What are the main claims of the paper and how significant are they for the discipline?

The authors investigate transcription dynamics in live cells using the well-established technique of labelling nascent mRNA molecules using the MS2/PP7 system. The main contribution of the paper is a deterministic model that describes transcription in terms of a few fundamental kinetic parameters. Despite its simplicity, the calibrated model is able to predict measured fluorescence levels fairly well. This is demonstrated on experimental data from the hunchback gene in the Drosophila embryo, which is known to show spatial variability of transcription levels.

%%%% Are these claims novel? If not, which published articles weaken the claims of originality of this one?

In my opinion the authors make too strong claims regarding the novelty of their general approach of using inference techniques in single-cell studies. I am aware that statistically sound inference is rarely done in the physics community (where it is usually degraded to a fitting procedure) but there is ample of work from the statistics and machine learning community to develop dedicated and sound inference schemes for single-cell data (e.g. papers of Finkenstadt and Rand, D. Suter et al. Science 2011, Zechner et al. Nature Methods 2014 and many more).

For parameter inference, a semi-Bayesian perspective is adopted. In particular, an individual Markov chain is run for each cell. The posterior samples are used to estimate the posterior mean per parameter per cell. Then, the authors use descriptive statistics on these MAP estimates. As MAP estimates are usually rather sensitive (in particular for sampling-based approaches) I am a bit concerned whether the weak correlations they are finding and interpreting are really significant. A more rigorous approach would have been to work with the full posterior distribution. From a computational perspective, the Monte Carlo inference procedure is quite standard. Considering this, the promise of a “novel computational technique to simultaneously infer […] parameters” as mentioned in the abstract may be a bit over the top. Nevertheless, the demonstrated results such as the spatial variation of the initiation rate are interesting. In my opinion, the most compelling methodological result is the use of the dual reporter system to eliminate the need for GFP calibration experiments. This is important, since GFP calibration has been a major drawback for model-based approaches in this area so far.

%%%% Are the claims properly placed in the context of the previous literature? Have the authors treated the literature fairly?

Related work is not fully captured (see above) although the list of references is rather extensive with respect to nascent mRNA labeling work. What I found surprising though, is that auto-correlation analysis (e.g. applied by Larson and co-workers) was not discussed at all. To my knowledge, this is a standard technique to analyze live cell transcription traces. I would have expected a discussion of the advantages and disadvantages of the proposed method compared to ACA.

%%%% Do the data and analyses fully support the claims? If not, what other evidence is required?

The proposed inference scheme relies on an established Monte Carlo procedure and seems to work fairly well. However, it may be helpful to refine the observation model given in (S12), (S13). The given likelihood function implicitly assumes that given the true intensity, the observations are distributed with a standard deviation of one around the true value. Such a small observation noise is not realistic for fluorescence measurements. I would propose to consider a multiplicative noise model (larger noise for larger intensity) or at least some relative error in (S13). This may also help with the problem that the tails gets too much emphasis during the fit as discussed in S3.2.

In my opinion, the hierarchical procedure proposed in S3.2 is a bit of a hack. There are more transparent methods to solve this problem, such as alternative observation models or using weighted residuals. The motivation to call some data points less important always seemed to be rooted in the bad fit they generate if considered full. A more sound way to model such discounting is through introducing heteroscedasticity in the observation model but that then still requires biophysical justification.

Testing the inference procedure on simulated results (S3.4) is helpful. I am surprised, though, that the inference result is so biased (Fig. S4 a). Considering that the model is relatively simple and the number of data points is large, I would not have expected such a mismatch. For me, this indicates some problem with the inference procedure. For example, the posterior could be multi-modal and the chain could be trapped in a local mode. The normalized error measure does not help to resolve this discrepancy.

In my opinion, more evidence is required for section S4. This part is based on a construct with 24 alternating MS2-PP7 stem-loops. This model has been established first using sm-FISH using FISH probes (Zoller et al.,2018). Compared with the MS2-PP7 system, the FISH probes can be labeled with a lot of fluorophores, so it is more convenient to measure fluorescence intensity. By using the MS2-PP7 system, normally more than 14 consecutive stem-loops should be used to generate a single spot. It would be helpful to provide more figures or videos in this section. Also, the photobleaching needs to be considered by using this system. It would be better to mention more details about the DNA construct in this part as well. The measurement condition should be mentioned in figure 5S.

The most problematic aspect of the paper seems to be the simulation study in S9. The idea is to support the claim of individual RNAPs having different step sizes by consulting a more elaborate simulation model of the transcription process. I see, however, some severe problems with the taken approach. First, the simulation itself is not very meaningful. Essentially, the result is that randomized RNAP step sizes produce random elongation rates, which seem quite trivial. Second, the authors compare this distribution of elongation rates to the distribution of inferred elongation rates. Conceptually, it does not make much sense to compare the distribution of inferred quantities with the distribution of a completely different generative process. What the authors could have done instead is to extend the fine-grained model in such a way that it can produce artificial observations. The artificial observation could then be used for inference as in S3.4. The obtained distribution of posterior means would allow an appropriate comparison with the distribution of inferred elongation rates from the real data.

The filtering of data-points for the synthetic data scenario is beyond justification. If the data was generated according to the model that is later used inference, every datapoint needs to be taken into account.

The aggressive down selection performed on the real dataset appears also very problematic. From the original 1053 cells after successive filtering only 299 remain in the inference dataset. For some data points, the only justification to discard them is that they cannot be well explained by model. In my opinion that is an elementary statistical fallacy.

In section DNA construct, they mentioned the paper Garcia et al.,2013. They used the almost the same DNA construct. However compared with the paper, it showed no background signal inside the cell. More details should be discussed in both DNA construct parts and also behind the figure 2A.

In section S7 (line 1130), figure S5C was used to explain the separation of the experimental noise from the biological noise. However the figure showed the fluorescence intensity of MS2/PP7 and a linear fit.

%%%% Are original data deposited in appropriate repositories and accession/version numbers provided for genes, proteins, mutants, diseases, etc.?

In section S4, a construct with alternating MS2/PP7 loops was used to calibrate the signals. The DNA construct is required.

%%%% Are details of the methodology sufficient to allow the experiments to be reproduced?

More details are required in the methods and sample preparation section.

In section sample preparation, only the reference papers were mentioned, the whole preparation process should be mentioned too.

In section image analysis, a custom-written software was used to analyze the images. The name and purpose should also be mentioned.

%%%% Is any software created by the authors freely available?

Github repository

%%%% Minor remarks regarding the modelling part

The high RNAP density (Fig. 3 d) seems to contradict the independent particle assumption of the model. Can you clarify why it is legitimate to still use this model?

The initiation rate R(t) in the description of the full model in S1 is not fully clear to me. The notation suggests that delta R(t) is a stochastic fluctuation, but from the description in the inference part, it seems like it is treated as a constant offset for each time point. Also, the discussion suggests that the model is in continuous-time. Then, in line 721, a computational time step suddenly appears. This could be explained more explicitly.

From (S4), (S5) it seems that the number of RNAP molecules is a discrete quantity. In contrast, the discussion in S1 around line 720 explains that R(t) dt RNAP molecules are loaded at each time step, which is not an integer.

Is there a particular reason mu_x is used in (S16) for the normalization? Why not use x_true as for a standard relative error?

**Have all data underlying the figures and results presented in the manuscript been provided?**

Reviewer #1: Yes

Reviewer #2: Yes

Reviewer #3: Yes

Reviewer #4: **No: **to my knowledge the raw time series data is not made available

PLOS authors have the option to publish the peer review history of their article (what does this mean?). If published, this will include your full peer review and any attached files.

Reviewer #1: **Yes: **Robin E. C. Lee

Reviewer #2: No

Reviewer #3: No

Reviewer #4: No
---

## [Decision Letter · Decision Letter 1]

10 Mar 2021

Dear Mr. Liu,

Thank you very much for submitting your manuscript "Real-time single-cell characterization of the eukaryotic transcription cycle reveals correlations between RNA initiation, elongation, and cleavage" for consideration at PLOS Computational Biology. As with all papers reviewed by the journal, your manuscript was reviewed by members of the editorial board and by several independent reviewers. The reviewers appreciated the attention to an important topic. Based on the reviews, we are likely to accept this manuscript for publication, providing that you modify the manuscript according to the review recommendations.

Sincerely,

James R. Faeder

Associate Editor

PLOS Computational Biology

Jian Ma

Deputy Editor

PLOS Computational Biology

[LINK]

Reviewer's Responses to Questions

**Comments to the Authors:**

Reviewer #1: In their revised manuscript, the authors have made several improvements through additional analysis and formatting. In particular, the revised work implements an improved observation model with a scaling noise term to relate intensity measurements for different fluorophores. The resulting data more consistently represents the underlying biological phenomena, averting the need for a hierarchical fitting procedure which came across as a ‘brute-force’ solution in the prior submission. The authors also implement an automated data curation method that removes human bias and preserves a greater number of single-cell trajectories. Overall the revised manuscript seems more substantive and scientifically sound with these changes in combination with wise formatting decisions. Although I still have a couple minor comments, I am satisfied by the author’s responses to my concerns in addition to those of my colleagues.

Minor comments:

1. My previous concern #1 related to assumptions of processivity and instantaneous cleavage of mRNA given the observation that mCherry fluorescence intensity can peak and begin decaying while the EGFP channel is still approaching its peak in the same cell. If I understand the response correctly, the argument is that the steady state plateau where mRNA cleavage and production rates are balanced ends earlier for the 5’ reporter than the 3’ reporter because elongation continues after transcript initiation has ceased. So in this intermediate state there is continued production of the 3’ reporter and no new production of the 5’ reporter, and the intensity of the 5’ reporter will therefore begin decaying because of ongoing cleavage events. Modifications to figure 1D, in particular the phase marked (vi), clearly indicate this process but I did not find an accompanying explanation in the manuscript or figure legend.

2. I will push back on the claim “... the computational algorithm itself is quite standard, but the application of Bayesian inference to directly fitting live imaging datasets is novel.” and draw the author’s attention to a couple our recent works PMID: 30175326 and PMID: 32150537. The first of which implements a Bayesian analysis technique to increase global sampling (i.e. preventing capture in local minima) and accelerates convergence for time-course data. The second implements a more advanced Bayesian approach and applies it to live-cell imaging data. These topics may be relevant to the discussion on ‘comparison to existing analysis techniques’ or the ‘future improvements’.

Reviewer #2: I like to thank the authors for responding to my critique in such detail. After reading the response carefully, I am mostly satisfied with the response of the authors. I disagree with the only point related to statements about the limitations of using RNA-FISH data to model gene expression dynamics from fixed cells. Specifically, the information in lines 454-455 is not correct. RNA-FISH can infer models and rates at a high temporal resolution down to 1min and at similar time scales as done so in this manuscript. What RNA-FISH cannot do is following the same single cell over time. An excellent example of where live cell transcription imaging is gaining novel insight compared to RNA-FISH is understanding mRNA – lncRNA regulation, for example, by work from the Larson lab [1]. The authors' lack of literature review might be their limited knowledge about the work by several groups that successfully used time course snapshot RNA-FISH data of fixed cells to infer rates and mechanisms of transcription similar as done in the current manuscript [2,3,12,13,4–11]. To place the authors' orthogonal work into perspective, I recommend including these and equivalent citations in the document. Specifically, add appropriate sources to the revised manuscripts in lines 96, 259, 319, 423, 455, Sections S5, S8, S10. I believe that live cell and fixed cell experiments of transcription complement each other and demonstrate overlapping conclusions. After this point has been addressed, I recommend the manuscript be published.

Bibliography:

1. Lenstra TL, Coulon A, Chow CC, Larson DR (2015) Single-Molecule Imaging Reveals a Switch between Spurious and Functional ncRNA Transcription. Mol Cell 60: 597–610.

2. Wyart M, Botstein D, Wingreen NS (2010) Evaluating gene expression dynamics using pairwise RNA fish data. PLoS Comput Biol 6: 1000979.

3. Gómez-Schiavon M, Chen LF, West AE, Buchler NE (2017) BayFish: Bayesian inference of transcription dynamics from population snapshots single-molecule RNA FISH in single cells. Genome Biol 18: 164.

4. Lyon K, Aguilera LU, Morisaki T, Munsky B, Stasevich TJ (2019) Live-Cell Single RNA Imaging Reveals Bursts of Translational Frameshifting. Mol Cell 75: 172-183.e9.

5. Xu H, Sepúlveda LA, Figard L, Sokac AM, Golding I (2015) Combining protein and mRNA quantification to decipher transcriptional regulation. Nat Methods 12: 739–742.

6. Neuert G, Munsky B, Tan RZ, Teytelman L, Khammash M, van Oudenaarden A (2013) Systematic Identification of Signal-Activated Stochastic Gene Regulation. Science (80- ) 339: 584–587.

7. Munsky B, Li G, Fox ZR, Shepherd DP, Neuert G (2018) Distribution shapes govern the discovery of predictive models for gene regulation. Proc Natl Acad Sci 115: 7533–7538.

8. Miura M, Dey S, Ramanayake S, Singh A, Rueda DS, Bangham CRM (2019) Kinetics of HTLV-1 reactivation from latency quantified by single-molecule RNA FISH and stochastic modelling. PLOS Pathog 15: e1008164.

9. Fei J, Singh D, Zhang Q, Park S, Balasubramanian D, Golding I, Vanderpool CK, Ha T (2015) Determination of in vivo target search kinetics of regulatory noncoding RNA. Science (80- ) 347: 1371–1374.

10. So L, Ghosh A, Zong C, Sepúlveda LA, Segev R, Golding I (2011) General properties of transcriptional time series in Escherichia coli. Nat Genet 43: 554–560.

11. Shaffer SM, Dunagin MC, Torborg SR, Torre EA, Emert B, Krepler C, Beqiri M, Sproesser K, Brafford PA, Xiao M, et al. (2017) Rare cell variability and drug-induced reprogramming as a mode of cancer drug resistance. Nature 546: 431–435.

12. Senecal A, Munsky B, Proux F, Ly N, Braye FE, Zimmer C, Mueller F, Darzacq X (2014) Transcription Factors Modulate c-Fos Transcriptional Bursts. Cell Rep 8: 75–83.

13. Albayrak C, Jordi CA, Zechner C, Lin J, Bichsel CA, Khammash M, Tay S (2016) Digital Quantification of Proteins and mRNA in Single Mammalian Cells. Mol Cell 61: 914–924.

Reviewer #3: The revised manuscript by Liu et al. has addressed the previous issues with model presentation, data curation and benchmarking of the inference method. The abstract, introduction and discussion have been adjusted to properly reflex the method itself and its applicability, rather than focusing too much on the biological findings. By applying a model of transcription to predict the MS2 traces, with the variance scaling with the mean signal as suggested by reviewer #4, the inferred variance from the data is found to be much higher than the inference error (Fig. S5D). Thus, conclusions on the variations and correlations between parameters are valid.

I only have some comments on this notion of scaled variance. They require very minor discussions but are important. Other than that, I find the revision satisfactory for publication.

-From Fig. S4, the found variance not only scales linearly but is proportional to the mean intensity. This implies that noise emerges purely from the GFP and mCherry molecules bound to nascent RNA, rather than the background noise (i.e. from unbound fluorescent molecules). Normally I would expect a mix of both, especially when using a gaussian filter to extract the MS2/PP7 spot intensity at the detected spot location (as in Garcia et al, 2013, Lucas et al, 2018). In this work, does the calculation of the spot intensity involve such a filter? Or is the spot intensity just the sum of detected spot’s pixel intensity, and thus the variance scales with number of detected pixels (i.e. spot size). Please discuss whether the scaling of signal variance can depend on how spot intensity is calculated, which is not always standardized.

-Given source of noise in the detected ms2 signal mostly arise from the nascent RNA (noise scaled with loci intensity) rather than from the background intensity (noise unscaled), can you discuss the viability of previous “ensemble” methods, such as memory based HMM or Autocorrelation analysis, which assume only background noise?

-Please use p-val or p-value instead of p, since it is confusing when placed next to rho sign.

-Line 101. Closing bracket needed

-Line 986. Please continue from “…”

**Have all data underlying the figures and results presented in the manuscript been provided?**

Reviewer #1: None

Reviewer #2: Yes

Reviewer #3: Yes

PLOS authors have the option to publish the peer review history of their article (what does this mean?). If published, this will include your full peer review and any attached files.

Reviewer #1: **Yes: **Robin E. C. Lee

Reviewer #2: No

Reviewer #3: No

Figure Files:

Data Requirements:

Reproducibility:

References:

---

## [Editor Report · Decision Letter 2]

23 Apr 2021

Dear Mr. Liu,

We are pleased to inform you that your manuscript 'Real-time single-cell characterization of the eukaryotic transcription cycle reveals correlations between RNA initiation, elongation, and cleavage' has been provisionally accepted for publication in PLOS Computational Biology.

Best regards,

James R. Faeder

Associate Editor

PLOS Computational Biology

Jian Ma

Deputy Editor

PLOS Computational Biology

---

## [Editor Report · Acceptance letter]

14 May 2021

PCOMPBIOL-D-20-01950R2 

Real-time single-cell characterization of the eukaryotic transcription cycle reveals correlations between RNA initiation, elongation, and cleavage

Dear Dr Garcia,

I am pleased to inform you that your manuscript has been formally accepted for publication in PLOS Computational Biology. Your manuscript is now with our production department and you will be notified of the publication date in due course.

With kind regards,

Zsofi Zombor
